# Firocoxib as a Potential Neoadjuvant Treatment in Canine Patients with Triple-Negative Mammary Gland Tumors

**DOI:** 10.3390/ani13010060

**Published:** 2022-12-23

**Authors:** Andressa Brandi, Patricia de Faria Lainetti, Fabiana Elias, Marcela Marcondes Pinto Rodrigues, Livia Fagundes Moraes, Renée Laufer-Amorim, Laíza Sartori de Camargo, Cristina de Oliveira Massoco Salles Gomes, Carlos Eduardo Fonseca-Alves

**Affiliations:** 1Department of Veterinary Surgery and Animal Reproduction, São Paulo State University-UNESP, Botucatu 18618-970, Brazil; 2Department of Veterinary Medicine and Animal Science, University of Milan, Via dell’ Università 6, 26900 Lodi, Italy; 3Veterinary School, Federal University of Fronteira Sul-UFFS, Realeza 85770-000, Brazil; 4CEVEPAT Veterinary Diagnostic Laboratory, Botucatu 18610-034, Brazil; 5Department of Veterinary Clinic, Sao Paulo State University-UNESP, Botucatu 18618-681, Brazil; 6Department of Pathology, University of São Paulo-USP, São Paulo 05508-270, Brazil; 7Institute of Health Sciences, Paulista University-UNIP, Bauru 17048-290, Brazil

**Keywords:** cyclooxygenase-2, caspase-3, chemotherapy, nonsteroidal anti-inflammatory drug

## Abstract

**Simple Summary:**

Mammary gland tumors are most frequent in non-castrated female dogs and represent an interesting model for understanding tumor biology of human breast cancer. Ciclooxigenase-2 (COX-2) is an enzyme associated with the inflammatory process and different studies have implied the association of COX-2 expression with patient’s prognosis. However, few studies have investigated the effects of drugs that block COX-2 enzyme. In this context, we performed in vitro and in vivo experiments to understand the effect of COX-2 blockers on mammary cancer cells that overexpress COX-2. We found an induction of mammary cancer cell apoptosis after treatment with COX-2 inhibitors indicating the role of COX-2 enzyme in mammary cancer cell maintenance. Moreover, COX-2 inhibitors can represent therapeutic effects on triple-negative mammary cancer.

**Abstract:**

This study aimed to investigate the pro-apoptotic effects of NSAID (Previcox^®^) in vitro and in vivo. Two CMT cell lines, one from the primary tumor and one from bone metastasis, were treated with firocoxib and MTT assay was performed to determine the half-maximal inhibitory concentration (IC_50_) value. The firocoxib IC_50_ for the cell lines UNESP-CM5 and UNESP-MM1 were 25.21 µM and 27.41 µM, respectively. The cell lines were then treated with the respective firocoxib IC_50_ concentrations and annexin V/propidium iodide (PI) assay was performed, to detect the induction of apoptosis in both cells (Annexin+/PI+). We conducted an in vivo study involving female dogs affected by CMT and divided them into control and treatment groups. For both groups, a biopsy was performed on day 0 (D0) and a mastectomy was performed on day 14 (D14). In the treatment group, after biopsy on D0, the patients received Previcox^®^ 5 mg/kg PO once a day until mastectomy was performed on D14. COX-2/caspase-3 double immunostaining was performed on samples from D0 and D14, revealing no difference in the control group. In contrast, in the treatment group Previcox^®^ increased the number of COX-2 positive apoptotic cells. Therefore, firocoxib can induce apoptosis in CMT cells in vitro and in vivo, and Previcox^®^ can be a potential neoadjuvant treatment for patients with mammary cancer.

## 1. Introduction

Breast cancer is one of the most common cancers in women worldwide, and the last global estimate expected more than 2 million new cases in 2018 [1]. Breast cancer in women is divided into HER2-enriched, luminal A, luminal B, and basal-like subtype [2]. The basal-like tumor group is also called triple-negative breast cancer (TNBC) because these tumors usually lack estrogen receptor (ER), progesterone receptor (PR), and human epidermal growth factor receptor 2 (HER2) expression [2]. TNBC is more aggressive than other breast cancer subtypes and represents 10–20% of breast cancer diagnoses in human patients [3,4]. Traditional prognostic predictors, such as tumor size and nodal status, are not precise for overall survival prediction in TNBC [3,5]. This tumor subgroup is therapeutically challenging, and patients are usually treated with radiation therapy and chemotherapy.

Mammary gland tumors are the most commonly diagnosed tumors in non-spayed female dogs and represent an important model for human breast cancer [6,7,8]. As canine mammary gland tumors develop spontaneously and are related to aging and hormonal imbalance, dogs can be an interesting model for comparative studies [9,10,11]. The triple-negative tumor subgroup seems to be frequent in veterinary medicine, and few previous studies have used this classification to associate clinical or pathological features with different tumor subtypes [6,7,8]. A recent study evaluating a large number of samples, dogs with triple-negative canine mammary tumors (TNCMTs) experienced a shorter survival time than those with luminal A and luminal B tumors [6]. These authors also demonstrated a high incidence of triple-negative tumors in female dogs, which is interesting from a comparative oncology perspective.

In both humans and dogs, cyclooxygenase-2 (COX-2) expression is associated with more aggressive tumor subtypes. Few studies have evaluated the association between TNBC and COX-2 expression in humans [12], and no previous study has evaluated the association between triple-negative tumors and COX-2 expression. COX-2 inhibitors have been claimed to have antitumor effects mostly based on in vitro studies [13]. However, there is limited evidence from in vivo studies [13,14,15,16,17]. Although COX-2 inhibitors have been claimed as potential treatment for female dogs affected by mammary gland tumors, no previous studies have described the potential antitumor effect of these drugs. Among the potential antitumor effects of COX-2 inhibitors, it is proposed the antiangiogenic potential, the proliferative effect, and induction of cancer cell apoptosis [12]. COX-2 is considered a membrane-bound enzyme, responsible for the inflammatory cascade, playing an important role on the synthesis of several inflammatory mediators such as prostaglandins and thromboxane [7].

In dogs, some studies have investigated the role of COX-2 in patient’s prognosis; however, no prospective studies investigating the COX-2 antitumor effect on canine mammary gland tumors were found. Therefore, this study aimed to evaluate the pro apoptotic effect of firocoxib on two different canine mammary gland tumor cell lines and on female dogs naturally affected by triple-negative mammary gland tumors.

## 2. Materials and Methods

### 2.1. Cell Lines

Two triple-negative canine mammary gland tumor cell lines, one from a triple-negative mammary gland tumor and the other from bone metastasis, and from a previously characterized TNCMT (UNESP-CM5 and UNESP-MM1) [9], were used in this study. Both the cell lines tested negative for mycoplasma contamination. Paraffin blocks from tumors that originated from both cell lines were also retrieved, and immunofluorescence for COX-2 was performed on both tumors according to Costa et al. [18]. To confirm COX-2 expression in UNESP-CM5 and UNESP-CM1 cells, COX-2 immunofluorescence (IF) was performed, as described by Kobayashi et al. [19]. Briefly, for IF assay, the primary polyclonal goat anti-human COX-2 antibody (Santa Cruz Biotechnology, Dallas, TX, USA) was used. Tissue samples were submitted to antigen retrieval using citrate pH 6.0 buffer on a commercial pressure cocker (Dako, Carpinteria, CA, USA) and the tissue samples were incubated with primary antibody at 1:100 dilution, overnight. Then, the slides were washed and incubated and Alexa Fluor 594 (BioLegend, San Diego, CA, USA) was applied at a 1.5 μg/mL dilution in PBS for 60 min at room temperature. The slides were counterstained with 4′-6-diamidino-2-phenylindole (DAPI; Sigma, Portland, OR, USA) and evaluated under a laser scanning confocal microscope (Leica Biosystems, Wetzlar, Germany).

### 2.2. Evaluation Cell Line Metabolic Activity and IC_50_ after Firocoxib Treatment

Evaluation of metabolic activity after firocoxib treatment and determination of half-maximal inhibitory concentration (IC_50_) were performed using an assay based on the cleavage of a tetrazolium yellow salt MTT [3-(4,5-dimethylthiazol-2-yl)-2,5-diphenyl tetrazolium bromide], according to the previous literature [20]. The 10^4^ of UNESP-CM5 and UNESP-MM1 cell lines were seeded in a 96-well plate containing Dulbecco’s Modified Eagle Medium/Nutrient Mixture F-12 (DMEM:F12) (Lonza, Basel, Switzerland) supplemented with 10% fetal bovine serum (FBS) (Lonza, Basel, Switzerland) and 1% penicillin and streptomycin mixture (Lonza, Basel, Switzerland), and incubated for 24 h in a humid atmosphere at 37 °C, before addition of firocoxib to the medium. The cells were then cultured and incubated in medium without FBS and treated with 1 µM, 5 µM, 10 µM, 50 µM, 100 µM, 500 µM, and 1000 µM concentrations of firocoxib for 24 h. Each experiment was performed in triplicate, with three plates per experiment (3 × 3). After incubation, 100 μL of MTT labeling reagent was added to each well, and the plate was incubated at 37 °C for 4 h. After incubation with MTT, the salt was solubilized with DMSO and the spectrophotometric absorbance of the samples was measured using a microtiter plate reader (Thermofisher, Waltham, MA, USA) at 570 nm. The IC_50_ values for each cell line were determined based on the previous literature [21]. The experiments were performed in biological and technical triplicate.

### 2.3. Annexin V/Propidium Iodide (PI) Apoptosis Assay

To evaluate necrosis and apoptosis induced by firocoxib IC_50_ treatment, the annexin V/propidium iodide (PI) apoptosis assay (Sigma-Aldrich, Saint Louis, MO, USA) was performed using flow cytometry, as described by Crowley et al. [22]. Briefly, cell lines were treated with a firocoxib IC_50_ dosage for 24 h, and the cells were harvested and counted. Subsequently, 1 × 10^5^ cells were added in 200 µL of Annexin V binding buffer containing 4 µL of 0.5 mg/mL PI and 2 µL of Annexin V-FITC and incubated for 15 min at room temperature in the dark. The cells were then subjected to flow cytometry according to the procedure described by Crowley et al. [22]. As a negative control, a cell line treated only with DMSO (diluent for firocoxib) was used. To interpret the results, viable cells were considered annexin V and PI negative, cells expressing annexin V and negative for PI were considered early apoptotic, cells positive for both annexin V and PI were considered late apoptotic, and cells negative for annexin V and positive for PI were considered necrotic [22].

### 2.4. In Vivo Experiment

All procedures performed in this study were in accordance with national and international guidelines for the use of animals in research. The in vivo experiments included client-owned dogs referred to the São Paulo State University Teaching Hospital between January and December 2017. The procedures were approved by the Institutional Ethics Committee of Animal use in Research (CEUA #0208/2016) and the owners signed an informed consent allowing the use of the obtained samples in research.

Twenty-five female dogs with triple-negative mammary gland tumors were randomized into two groups: control (*n* = 12) and treatment (*n* = 13). In both groups, an incisional biopsy was performed for histopathological diagnosis (D0). After incisional biopsy, all patients (*n* = 25) were treated with Firocoxib for three days, followed by four days free of any treatment. At day seven (D7), all patients had no sign of local inflammation and the patients from the control group received only necessary medical treatment and those from the treatment group received firocoxib at 5 mg/kg/day, orally for seven days (completing a total of 14 days after initial biopsy (D14)). After this period (D14), all the dogs underwent radical unilateral mastectomy and histopathological analysis. Therefore, tissue samples for immunohistochemistry were achieved from incisional biopsy at the D0 and total mastectomy on D14. Histopathological evaluation was performed according to Goldsmith et al. [23], and tumor grading was performed according to Karayannopoulou et al. [24]. The molecular phenotypes were evaluated as described by Abadie et al. [6]. Although this was not the goal of this study, the Response Evaluation Criteria for Solid Tumors in dogs was applied to assess the tumor volume [25]. Tumor size was measured using a digital caliper.

### 2.5. Inclusion and Exclusion Criteria

Only subjects with mammary gland carcinomas presenting epithelial malignant proliferation (simple subtype) and no mesenchymal proliferation (benign or malignant) were included in this study. Owing to mammary gland tumor heterogeneity, subjects with discrepancies between histopathological analyses (D0 and D14) were excluded from this study. Subjects with comorbidities and those who had received previous anti-inflammatory treatments were excluded. Side effects were evaluated and graded according to the Veterinary Cooperative Oncology Group-Common Terminology Criteria for Adverse Events recommendations [26].

### 2.6. COX-2/Cleaved Caspase-3 Double Immunostaining

The protocol for COX-2/cleaved caspase-3 double staining was performed according to a previous study [27]. Double staining analysis was performed on tumor samples by comparing D0 and D14 for each patient. Briefly, a 2-color double immunostaining was performed using COX-2 (DAB, brown staining) and cleaved caspase-3 (AEC, red staining). Antigen retrieval was performed using citrate solution (pH 6.0) in a pressure chamber (Pascal, Dako, Carpinteria, CA, USA) for 30 s at 125 °C, and then for 25 min at 94 °C. Endogenous peroxidase was blocked using a commercial solution (Dako, Carpinteria, CA, USA) for 10 min, and non-specific protein binding was blocked with a commercial solution (Protein block^®^, Dako, Carpinteria, CA, USA) for 30 min. A polyclonal anti-human COX-2 antibody (Santa Cruz Biotechnology, Dallas, TX, USA) was then applied at a 1:100 dilution overnight. Then, a secondary antibody was applied for 60 min (Envision, Dako, Carpinteria, CA, USA) and 3 3′-diaminobenzidine was used as the chromogen. The second primary antibody applied was rabbit monoclonal anti-human cleaved caspase-3 (Cell Signaling, Danvers, MA, USA) at 1:300 dilution overnight [28]. Then, a polymer system was used as a secondary antibody for 60 min (Envision, Dako, Carpinteria, CA USA), and the 3-amino-9-ethylcarbazole chromogen (AEC) (Dako, Carpinteria, CA, USA) was applied. The slides were counterstained with Harris hematoxylin for 1 min and topped with a coverslip using an aqueous method (Dako, Carpinteria, CA, USA).

With respect to immunohistochemical interpretation, COX-2 antibody is expressed in the cell cytoplasm and cleaved caspase-3 in the nucleus. Therefore, brown cytoplasmic expression was considered to be COX-2 immunolocalization, red-nuclear staining, cleaved caspase-3 expression, and blue staining was interpreted as negative expression. Canine kidney and lymph node were used as positive control for COX-2 and cleaved caspase-3 expression, respectively. The negative control was performed using isotype immunoglobulin at the same concentration as the respective primary antibodies.

Regarding the cells analysis, the whole immunohistochemistry slide were evaluated and the hot spot areas were selected. Then a total of 1000 cells were counted and we used for analysis only the double-stained cells.

### 2.7. Statistical Analysis

The median tumor size was evaluated and the difference in tumor size between the treatment and control groups was compared using the Mann–Whitney U test. The mean value and standard deviation were used for descriptive analysis and the Mann–Whitney U test was used to evaluate the differences in the number of double-stained cells (COX-2/Caspase-3) between the control and treatment groups.

## 3. Results

### 3.1. Cell Line Characteristics

UNESP-CM5 is a cell line that originated from a tubulopapillary grade II mammary gland tumor with a triple-negative basal-like phenotype. COX-2 immunoexpression in the tissue samples revealed COX-2 expression in more than 75% of cancer cells (Figure 1A). The cultured cells showed fibroblastic to polygonal morphology in the culture flasks (Figure 1C), with COX-2 expression. The UNESP-MM1 cell line originated from a metastatic bone lesion of solid carcinoma with a triple-negative basal-like phenotype. The tissue sample that the cell line originated from showed COX-2 expression in more than 75% of the cancer cells (Figure 1B), and the cell line in the culture flask presented a polygonal morphology (Figure 1D). Culture of UNESP-MM1 cells also showed COX-2 expression in over than 50% of cancer cells with cytoplasmic expression.

### 3.2. Firocoxib-Induced Apoptosis in CMT Cell Lines

The UNESP-CM5 cell line was sensitive to firocoxib, showing an IC_50_ of 25.21 µM, and the UNESP-MM1 cell line had an IC_50_ of 27.41 µM (Figure 2A,B). The cell lines were treated with their respective IC_50_ concentrations and cellular apoptosis was evaluated. The UNESP-CM5 cell line had 33.1% of the cells positive for both annexin V and PI (cells in late apoptosis) and 45.9% of the cells negative for both markers. The UNESP-MM1 cell line had 74.9% of the cells positive for annexin V and PI (cells in late apoptosis) and 6.6% of the cells negative for annexin V and PI (Figure 2 and Table 1).

### 3.3. Firocoxib Increased Apoptosis in TNCMT

Twenty-five dogs were randomly included in this study, with 12 female dogs in the control group and 13 in the treatment group. The clinical and demographic information for each patient is shown in Table 2. The median tumor volume (in cm^3^) of the control and treatment groups at D0 were 20.4 (1–172.5) and 10.5 (1.16–160), respectively. After seven days of treatment, no significant changes were observed. Therefore, all the patients were considered to have stable disease.

None of the patients experienced side effects of Previcox^®^. The COX-2 expression did not show statistical difference compared with both groups at D0. Moreover, when compared with the COX-2 expression at D0 versus D14 in the treated group, no statistical difference was found (Appendix A). In the control group, the mean number of COX-2/caspase-3 double-stained cells in the biopsy sample was 7.9 ± 2.26 on D0 and 9.5 ± 2.08 on D14. In the treatment group, the mean number of COX-2/caspase-3 double-stained cells was 8.7 ± 2.36 prior treatment on D0 and 18.15 ± 4.62 after treatment on D14 (Figure 3). No statistical difference was observed in the number of double-stained cells in the control group between D0 and D14. However, in the group of patients treated with firocoxib, a higher number of apoptotic cells was observed after treatment with Previcox^®^ (P = 0.0007). Thus, firocoxib treatment increased the apoptosis of COX-2 positive cells. 

## 4. Discussion

COX-2 expression has been well-described in canine epithelial cancers, and COX-2 inhibitors have been proposed as a treatment option for COX-2 overexpression tumors [14]. However, most studies have evaluated only COX-2 expression in tissue samples (without treatment with COX-2 inhibitors), or COX-2 inhibitors are used in chemotherapeutic protocols with no assessment of COX-2 expression and its association with antitumor response [15,16,17,29,30,31]. Gregorio et al. [14] performed an interesting meta-analysis of COX-2 values as a prognostic factor and identified 272 published manuscripts evaluating COX-2 expression in canine and feline cancers. Among these studies, only 18 investigated COX-2 expression associated with the patients’ overall survival. Moreover, of the 18 selected studies, only three included COX-2 inhibitors in the patients’ treatment protocol [14]. Therefore, although COX-2 inhibitors have been claimed to exhibit antitumor effects, no previous studies have demonstrated the antitumor mechanism involved in clinical use of the inhibitors.

In the previous meta-analysis, seven studies have investigated the COX-2 prognostic value in CMT [14]. In the past studies, authors have evaluated COX-2 expression in canine CMT tissue samples, associating with overall survival, in a retrospective perspective. This is a very important point, since retrospective studies have a heterogeneous population and several variables can interfere in the results. According to Gregorio et al. [14], six previous studies in the veterinary literature provided sufficient evidence that COX-2 overexpression is associated with patient’s overall survival. However, these authors highlighted that only two studies have performed adjustments for variables of confusion [14]. Thus, the other four related studies can present their results secondary to other aspects. We performed a prospective study investigating the role apoptosis associated with a COX-2 inhibitor, using a strong inclusion and exclusion criteria, providing a strong evidence of the pro-apoptotic effect of Firocoxib on CMT cancer cells positive for COX-2.

On the other hand, our study has some limitations that we tried to assess and minimize prior the study conduction. The first limitation is related with the sample size. We used 25 female dogs divided into two groups. To address the limitation, we performed a power analysis to ensure the reliability of our study and it is important to highlighted that prospective clinical studies using this approach are very rare in the literature because of the need to perform tumor biopsy prior and after the treatment. The second important limitation is the heterogeneity of CMT. To minimize this heterogeneity, we included only triple-negative tumors. The last and important limitation was associated with the inflammation provided by tumor biopsy prior treatment, since it can induce tumor inflammation and increase COX-2 expression and cancer cell apoptosis. To address this limitation, both groups were submitted to tissue biopsy in the same moment and patients were treated with COX-2 inhibitors to avoid inflammation related to the biopsy.

Firocoxib is a COX-2 specific nonsteroidal anti-inflammatory drug previously tested in dogs, with minimal side effects. Previously, firocoxib was tested in geriatric dogs over a period of 90 days, with minimal biochemical changes and adverse drug events [29]. Therefore, it is not surprising that no side effects were observed in this study. Because we evaluated dogs for a shorter period, we used Previcox^®^ within its safety range. Considering that cancer occurs among older dogs, firocoxib is an excellent candidate as an adjuvant drug in cancer chemotherapy. However, Previcox^®^ is primarily prescribed as a pain reliever to treat inflammation associated with osteoarthritis, with no direct indication for treating dogs with cancer. Therefore, we performed a clinical study on female dogs with CMT. A tissue biopsy was performed on D0, the dogs were treated with Previcox^®^ at 5 mg/kg PO for seven days, and the patients were then subjected to radical mastectomy [30,31]. COX-2 expression was assessed on D0 and D14, and no statistical difference was found between the results from D0 and D14, indicating that Previcox^®^ did not decrease COX-2 expression in cancer cells in a short period of evaluation.

Our results indicate that COX-2 inhibitors did not decrease COX-2 expression in cancer cells in vivo in a short period of administration, raising some questions regarding in vivo COX-2 inhibition. We hypothesized that COX-2 inhibitors do not decrease COX-2 expression at the cellular level in cancer cells, but rather that COX-2 inhibition induces apoptosis in cancer cells positive for COX-2, and higher COX-2 expression could be associated with a higher proapoptotic effect. To confirm this hypothesis, we selected two CMT cancer cell lines originating from TNCMT and evaluated apoptosis in both of them after treatment with firocoxib at IC_50_. Our cell culture experiments confirmed the induction of apoptosis in both cell lines after firocoxib treatment, supporting the results of Previcox^®^ treatment in vivo.

On the other hand, we believe that a long-term use can induce COX-2 downregulation. In a previous case report published by our research group [32], we treated with Firocoxib for eight months a dog with intranasal carcinoma and we performed tumor biopsy prior and after treatment and assessed COX-2 expression in both moments. In this case report, we identified a reduction in COX-2 expression after eight months of continuous Firocoxib administration [32]. Therefore, in a long-term use, Fircoxib can induce COX-2 downregulation by cancer cells.

In vivo experiments did not reveal any statistical difference in the number of apoptotic cells on D0. However, in the samples from the treatment group on D14, we found a higher number of apoptotic cells positive for COX-2, indicating apoptosis secondary to Previcox^®^ treatment. Since patients treated with Previcox^®^ comprised a higher number of COX-2 positive cells with increased cleaved caspase-3 expression, we suggest that COX-2 inhibitors induce apoptosis in vivo. Therefore, patients with COX-2 positive tumors may benefit from treatment with Previcox^®^. Since surgery is the gold standard treatment for CMT, we recommend the use of Previcox^®^ for patients after surgery as an adjuvant treatment for COX-2 overexpression tumors or in the palliative treatment of COX-2 positive CMT.

The Firocoxib plasma concentration after 32 h of administration is 100 ng/mL [32]. Another recent study showed that Firocoxib plasma concentration range drastically in the first hours after administration, ranging to 100 up to 1000 ng/mL in the first hours. Therefore, our concentrations identified in the in vitro study, is in accordance with the maximum plasma concentration in vivo, allowing the use of the Firocoxib in the clinical practice for female dogs affected by CMT.

Apoptosis mediated by anti-inflammatory drugs is widely studied in human cancers [33,34,35,36,37,38], including colorectal cancer [34], ovarian cancer [35], hepatocellular carcinoma [36,37], and prostate [38]. The exact mechanism involved in apoptosis induced by COX-2 inhibitors in under debate [36,37]. However, it is well-established that COX-2 overexpression by cancer cells is associated with resistance to apoptosis. Thus, blocking COX-2 in cancer cells is directly associated with induction of apoptosis. Thus, in human cancers, association of anti-inflammatory drugs with maximum tolerance chemotherapy can enhance tumor apoptosis.

## 5. Conclusions

Overall, our results strongly suggest that firocoxib induces apoptosis in CMT cancer cells, and female dogs bearing COX-2 overexpressing tumors can benefit from Previcox^®^ treatment. Further studies including a higher number of patients and evaluating patient’s overall survival can increase the understanding of COX-2 inhibitors in cancer apoptosis.

## Figures and Tables

**Figure 1 animals-13-00060-f001:**
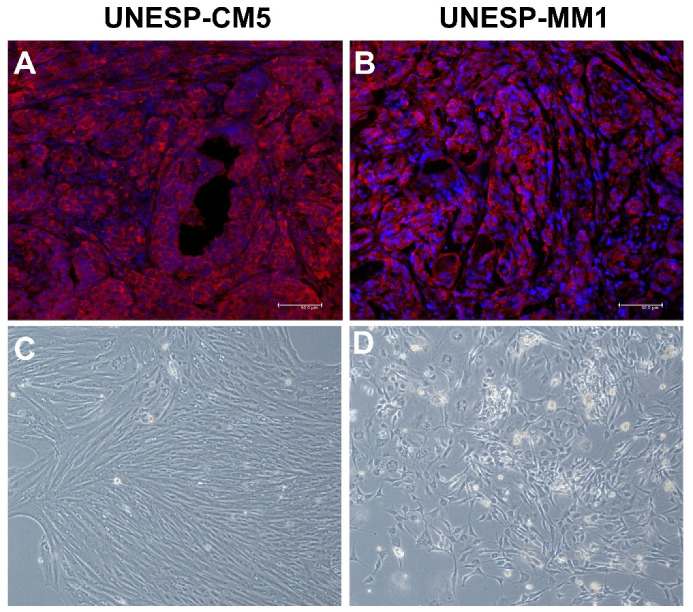
COX-2 expression and morphological characteristics of UNESP-CM5 and UNESP-MM1 from tumor samples and cancer cell lines. (**A**): primary tumor that caused the UNESP-CM5 cancer cell line presenting COX-2 positive immunofluorescence (red color). (**B**): tumor sample that caused UNESP-MM1 cell line presenting COX-2 overexpression (red color) by immunofluorescence. (**C**): fibroblastoid to polygonal morphology of UNESP-CM5 cell line (20×). (**D**): polygonal morphology of UNESP-MM1 cell line (20×).

**Figure 2 animals-13-00060-f002:**
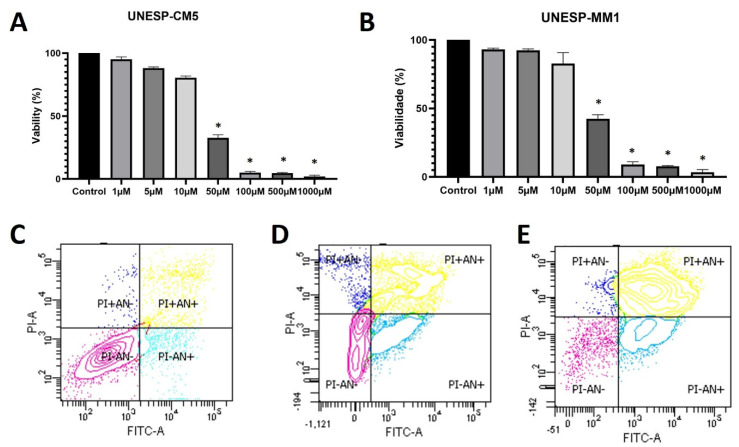
Cell culture experiments of UNESP-CM5 and UNESP-MM1 cancer cell lines. (**A**): cellular viability of canine UNESP-CM5 cell line under different firocoxib concentrations. (**B**): UNESP-MM1 cell line viability after treatment with different firocoxib concentrations. (**C**): Annexin V (AN) and propidium iodide (PI) assay in a pool of both cell lines (UNESP-CM5 and UNESP-MM1) without treatment as a negative control. The control cells presented more than 85% viability (PI−/AN−). (**D**): UNESP-CM5 cell line presenting 33.1% of the cells with PI+/AN+ and 45.9% of the cells negative for both. (**E**): UNESP-MM1 cell line showing more than 74% of the cells positive for PI and AN (PI+/AN+). The asterisk (*) means statistical difference compared with control group.

**Figure 3 animals-13-00060-f003:**
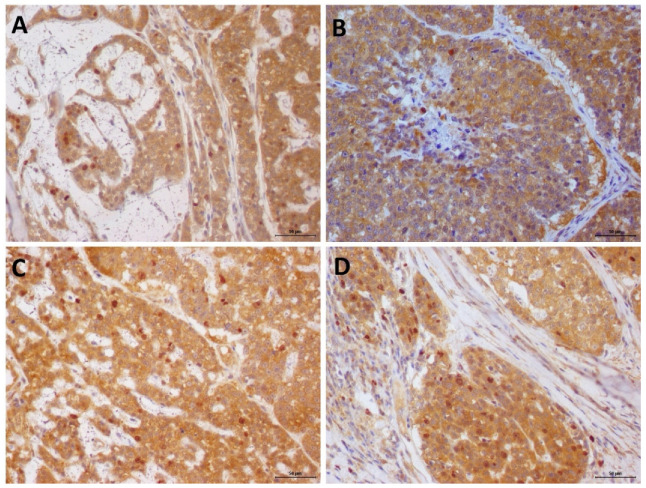
Double immunostaining of COX-2 (cytoplasmic staining—brown) and cleaved caspase-3 (nuclear staining—red) in canine mammary gland tumor samples. (**A**,**B**): COX-2/caspase-3 double immunostaining at D0. (**C**): COX-2/caspase-3 double staining of the control group tissue samples, on D14. No difference was found when the tissue samples from patients on D0 were compared with those on D14. (**D**): increased number of double-stained cells in a sample on D14, compared with those on D0. Harris hematoxylin counterstain, 3 3′-diaminobenzidine chromogen, and 20× magnification.

**Table 1 animals-13-00060-t001:** Evaluation of the number of cells positive for annexin V (AN) and propidium iodide (PI) in control, UNESP-CM5, and UNESP-MM1 groups.

Apoptosis Analysis	Control	UNESP-CM5	UNESP-MM1
**PI+/AN−**	0.8%	10.5%	3.4%
**PI+/AN+**	6.6%	33.1%	74.9%
**PI−/AN+**	85.9%	45.9%	6.6%
**PI−/AN−**	6.6%	10.5%	15.2%

Annexin V: AN; propidium iodide: PI; positive: +; negative: −; PI+/AN−: necrotic cells; PI+/AN+: late apoptosis; PI−/AN+: early apoptosis; PI−/AN−: viable cells.

**Table 2 animals-13-00060-t002:** Clinical information of the 25 canine patients used in the in vivo experiment.

Identification	Breed	Age (Years)	Weight	Tumor Volume (cm³)	Lymph Node Metastasis	Necrosis	Ulceration	Diagnosis	Grade	Overall Survival
*Control group*										
Case 1	MBD	12	18.2	28	Yes	No	No	Comedocarcinoma	Grade 3	35
Case2	MBD	11	14.7	172.5	No	Yes	Yes	Solid carcinoma	Grade 2	90
Case 3	MBD	11	15	108	Yes	No	Yes	Carcinoma in mixed tumor	Grade 2	365
Case 4	Sheep dog	9	28	15.75	No	No	No	Micropapillary	Grade 3	1095
Case 5	Rottweiler	10	27.2	156	No	No	No	Solid carcinoma	Grade 2	485
Case 6	Poodle	8	8.9	1.9	No	No	No	Tubulopapillary carcinoma	Grade 1	2531
Case 7	Beagle	9	18.3	1.6	No	No	No	Carcinoma in mixed tumor	Grade 2	1610
Case 8	Poodle	5	4.5	3	No	Yes	No	Complex carcinoma	Grade 2	90
Case 9	German Shepherd	7	31	35	No	Yes	No	Solid carcinoma	Grade 2	880
Case 10	Poodle	6	4.1	25	No	No	No	Tubulopapillary carcinoma	Grade 2	2737
Case 11	Saint Bernard dog	5	57	1	No	No	No	Carcinoma in mixed tumor	Grade 1	1095
Case 12	Poodle	5	3.8	1.25	No	No	No	Complex carcinoma	Grade 1	3650
*Treatment group*										
Case 13	Brazilian Bullmastiff	12	44	5.95	No	No	No	Adenosquamous carcinoma	-	425
Case 14	MBD	8	3.7	3	No	No	No	Carcinoma in mixed tumor	Grade 2	1670
Case 15	MBD	12	6.9	25	No	No	No	Solid carcinoma	Grade 2	180
Case 16	MBD	14	12	12	Yes	No	Yes	Anaplastic carcinoma	-	47
Case 17	Pinscher	12	2.6	10.5	No	No	No	Complex carcinoma	Grade 1	240
Case 18	Teckel	13	9.6	6	No	Yes	No	Complex carcinoma	Grade 1	1395
Case 19	MBD	8	7.5	3.08	No	No	No	Complex carcinoma	Grade 2	695
Case 20	Teckel	11	7.5	45.5	Yes	No	No	Solid carcinoma	Grade 2	395
Case 21	MBD	11	6.5	12.25	No	No	No	Tubulopapillary carcinoma	Grade 3	300
Case 22	Labrador	4	31.9	35	No	No	No	Complex carcinoma	Grade 2	1987
Case 23	MBD	10	34	160	No	No	Yes	Tubulopapillary carcinoma	Grade 2	455
Case 24	German Shepherd	7	18	3	No	No	Yes	Tubulopapillary carcinoma	Grade 2	270
Case 25	MBD	8	3.2	1.16	No	No	No	Complex carcinoma	Grade 1	1885

MBD: mixed breed dog.

## Data Availability

No new data were created or analyzed in this study. Data sharing is not applicable to this article.

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
