# Peer review of "Firocoxib as a Potential Neoadjuvant Treatment in Canine Patients with Triple-Negative Mammary Gland Tumors"

_animals, 2022, doi:10.3390/ani13010060_

Round 1

Reviewer 1 Report

The article demonstrates apoptosis induction associated with Cox2 inhibition treatment in vitro and provides relevant information to support the use of firocoxib in canine mammary cancer patients and its potential as adjuvant therapy.

Main comments:

1. There is a small number of clinical trials with different types of cancer in dogs (including mammary cancer) supporting the findings of the present work, however, those were not discussed in the present study. The authors mentioned several times (intro and discussion) that there is no evidence of anti-tumour effects in vivo. However, there is no mention of articles demonstrating otherwise. I would recommend the authors to expand the discussion on this topic besides the cited meta-analysis (https://doi.org/10.1002/vms3.460).

2. Furthermore, the discussion makes no reference to possible limitations of the present study (sample size, study duration, etc.).

3. There is also no discussion of the possible effects of long-term treatment on Cox 2 expression, would this possibly reduce the number of Cox2 + cells, and not only increase apoptosis?

4. Are there any suggestions for future research arising from the findings of the present study?

5. Are firocoxib concentrations for in vitro experiments below the range of the in vivo maximum plasma concentration in dogs treated with clinically relevant dosages? 

Specific comments:

L21: If by “important” you mean “common or frequent” better to use a more precise word as important might be ambiguous.

L22: represents.

L54: in human patients, or in humans.

L75: references

L84-87: Please provide a brief description of immunofluorescence methods (antibodies/clones). References are not enough to replicate your methods.

L88: Please correct this subtitle to clearly indicate that you are referring to the “cell lines” metabolic activity after firocoxib treatment, it doesn’t read well.

L99: are these concentrations below the range of the in vivo maximum plasma concentration in dogs treated with clinically relevant dosages?

L110: please correct 1×105

L103: plate reader details

L107: provide PI abbreviation

L119: please consider splitting this section into two

L126-133: Here is not clear what D0 and D7 means (before this point, it is only described in the abstract) maybe this whole paragraph might read better if placed after the next one, once the timeline has been described in the methods.

L136: maybe add: …during necessary medical treatment

L136-8: did treatment start directly after the biopsy or 7 days later? Is not quite clear. Describe biopsy methods (Incisional, trucut? Wedge?)

L172: estimated tumour size? Caliper-measured? Pathologic tumour size?

L179-187: please describe Cox-2 expression in cell lines. %, intensity, or distribution?

Fig 2: in Fig 2D please move PI+ AN- to avoid overlapping with the dot-plot. Please highlight significant differences between controls and different firocoxib concentrations in A and B with asterisks in the plot

Table 1: last two rows appear both as PI-/AN-, I guess the last one might be early apoptosis PI-/AN+? To facilitate table understanding please add a first column (vital early apoptotic, late apoptotic, necrotic) or describe PI/AN interpretation in the legend. The table should be understandable without reading the text

Table 2: are there any associations between clinical/HP/IHC variables and survival? Was this tested? If not why?

L225: I wouldn’t include NS p-value for controls, just mention it was not significant. Was this significant in the treatment group? (8.7±2.36 prior treatment on D0 Vs 18.15±4.62 at D7), you mentioned in the discussion (L267-268) this was not significant but it is not clear in the results section.                                                                                  

L245: correct ITS, please

L251-2: there are at least two clinical trials demonstrating improved survival in canine mammary cancer using long-term firocoxib as an adjuvant in comparison to chemotherapy, also a couple in bladder cancer and lymphoma. Please check other sources not considered in the meta-analysis

Author Response

Dear reviewer, thank you so much for your nice overview of our manuscript and for all positive criticism. After reading your comments, we identified several parts of our manuscript that were not clear, and we believe we had a strong improvement after including your comments. We kindly we would like to thank for your time reading our manuscript and for all comments provided. Please, see the specific answers below, and all modification are highlighted in red in the final manuscript.

There is a small number of clinical trials with different types of cancer in dogs (including mammary cancer) supporting the findings of the present work, however, those were not discussed in the present study. The authors mentioned several times (intro and discussion) that there is no evidence of anti-tumour effects in vivo. However, there is no mention of articles demonstrating otherwise. I would recommend the authors to expand the discussion on this topic besides the cited meta-analysis (https://doi.org/10.1002/vms3.460).

Answer: Thank you so much for your comments and we agree with reviewer’s opinion. Sorry, for not providing a more deep discussion. We have cited the mentioned meta-analysis (reference 25) in the fist version. However, we did not expand the discussion based on the meta-analysis. Therefore, we increased the discussion section, providing more specific information. 

Furthermore, the discussion makes no reference to possible limitations of the present study (sample size, study duration, etc.).

Answer: Thank you so much for this great suggestion and we apologize for not stating our limitations. Our major limitation are associated with the number of subjects included in the study, the heterogeneity of canine mammary gland tumors and the fact of the prior biopsy could induce inflammation in the tumor, increasing the expression of COX-2. Therefore, we have included in the discussion section a specific section for this limitations.

There is also no discussion of the possible effects of long-term treatment on Cox 2 expression, would this possibly reduce the number of Cox2 + cells, and not only increase apoptosis?

Answer: Thank you so much for you comment and it is a really interesting comment. We previously treated a canine nasal carcinoma with Firocoxib for eight months and we performed COX-2 expression in the tumor in the moment prior treatment and 8-months after treatment. In this case, we found a significant reduction of COX-2 expression in this patient. Regarding canine mammary tumors (CMT), they are usually treated with surgery. However, some tumors can show relapse and metastasis. In this scenario, we believe that COX-2 inhibitors could decrease COX-2 expression by cancer cells. We also have included a discussion about his point in discussion section.  

Are there any suggestions for future research arising from the findings of the present study?

Answer: Yes, as future suggestion, studies assessing the long-term apoptosis and reduction of COX-2 expression in female dogs with aggressive subtypes, associated with local relapse could be interesting. We have some unpublished data on cribriform tumors with local relapse from female cats, showing that long-term use of piroxicam associated with low dose of cyclophosphamide induce apoptosis, reduction in the tumor size and COX-2 expression. Therefore, the next step of this research should be repeating the same experiment with dogs with advanced cancer.

Are firocoxib concentrations for in vitro experiments below the range of the in vivomaximum plasma concentration in dogs treated with clinically relevant dosages

Answer: Yes, it was one of the important point of this study. We have included in the discussion section. According to the previous literature Firocoxib in a regular dosage reach 100 ng/ml in plasma. Therefore, our dosage in cell treatment were lower than maximum plasma concentration.

Specific comments:

L21: If by “important” you mean “common or frequent” better to use a more precise word as important might be ambiguous.

Answer: we have adjusted as suggested.

L22: represents.

Answer: we have adjusted as suggested.

L54: in human patients, or in humans.

Answer: we have included “in humans”

L75: references

Answer: we have included, as suggested.

L84-87: Please provide a brief description of immunofluorescence methods (antibodies/clones). References are not enough to replicate your methods.

Answer: we have adjusted as suggested.

L88: Please correct this subtitle to clearly indicate that you are referring to the “cell lines” metabolic activity after firocoxib treatment, it doesn’t read well.

Answer: we have adjusted as suggested.

L99: are these concentrations below the range of the in vivo maximum plasma concentration in dogs treated with clinically relevant dosages?

Answer: Yes, please, see the specific comment above and we also included in discussion section. 

L110: please correct 1×105

Answer: we have adjusted as suggested.

L103: plate reader details

Answer: we have adjusted as suggested.

L107: provide PI abbreviation

Answer: we have adjusted as suggested.

L119: please consider splitting this section into two

Answer: we have followed the suggestion and split into two section. 

L126-133: Here is not clear what D0 and D7 means (before this point, it is only described in the abstract) maybe this whole paragraph might read better if placed after the next one, once the timeline has been described in the methods.

Answer: We are so sorry for this confusion. Thank you for your comment and observation because the number were incorrect in our manuscript. The D0 was the time of first approach and biopsy. In the next seven days (D7) all patients were treated for three days with firocoxib to avoid inflammation related to the biopsy and the remain 4 days was used as a time free of Firocoxib. This was used for both groups. At the D7, the treated group received 7 days of Firocoxib and the control group was followed up and after 7 days (Total of 14 after first biopsy), mastectomy was performed. Therefore, we have tissue samples from two moments (D0 and D14). We have adjusted in the manuscript.

L136: maybe add: …during necessary medical treatment

Answer: we have modified as suggested.

L136-8: did treatment start directly after the biopsy or 7 days later? Is not quite clear. Describe biopsy methods (Incisional, trucut? Wedge?)

Answer: The first biopsy was incisional, followed for three days of firocoxib treatment and for days free of treatment. At the D7, the treated group started with Firocoxib for 7 days and after a total of 14 days from the first biopsy (7 days after firocoxib treatment), mastectomy was performed.

L172: estimated tumour size? Caliper-measured? Pathologic tumour size?

Answer: we have use a digital calliper following the Veterinary Cooperative Oncology Group recommendations.

L179-187: please describe Cox-2 expression in cell lines. %, intensity, or distribution?

Answer: we have included as suggested. 

Fig 2: in Fig 2D please move PI+ AN- to avoid overlapping with the dot-plot. Please highlight significant differences between controls and different firocoxib concentrations in A and B with asterisks in the plot.

Answer: We have included as asterisk to indicate statistical difference.  Regarding the caption of the image, our cytometer export a pdf as result and provide the legend (AN/PI) is not “changeable”.  We tried using different programs but the quality was very low. We are sorry, but we were not able to change. If the reviewer have any suggestion for programs that can change, we will be more than happy to change.

Table 1: last two rows appear both as PI-/AN-, I guess the last one might be early apoptosis PI-/AN+? To facilitate table understanding please add a first column (vital early apoptotic, late apoptotic, necrotic) or describe PI/AN interpretation in the legend. The table should be understandable without reading the text

Answer: we have included as suggested. 

Table 2: are there any associations between clinical/HP/IHC variables and survival? Was this tested? If not why?

Answer: Dear reviewer, we did not tried any of this associations. Our main manuscript focus is not associate COX-2 expression with survival. Our main focus is understand the mechanism involved in COX-2 antitumor effect. We have performed a power analysis to test if our number of samples it will be sufficient for a survival analysis, and the power test revealed a 100 patients as a minimum number for a reliable result. Therefore, providing this analysis it will not correlated with our data (only seven days of Firocoxib could not impact in overall survival), will not correlated with survival (25 patients is a number very low to identify any reliable result) and mammary gland tumours are very heterogeneous in morphological and clinical aspects. Therefore, using this analysis in only 25 patients will not be reliable.

L225: I wouldn’t include NS p-value for controls, just mention it was not significant. Was this significant in the treatment group? (8.7±2.36 prior treatment on D0 Vs 18.15±4.62 at D7), you mentioned in the discussion (L267-268) this was not significant but it is not clear in the results section.  

Answer: sorry for the confusion in this part. We re-write this part and we did not find difference when compared only COX-2 expression prior and after treatment. But in apoptosis we found difference. Therefore, we modified as suggested.                                                                                 

L245: correct ITS, please

 Answer: we have included as suggested. 

L251-2: there are at least two clinical trials demonstrating improved survival in canine mammary cancer using long-term firocoxib as an adjuvant in comparison to chemotherapy, also a couple in bladder cancer and lymphoma. Please check other sources not considered in the meta-analysis

Answer: we are so sorry for this confusion. We were meaning studies evaluating mechanism involved in COX-2 antitumor effect. Not overall survival. Therefore, we modified this part for a better clarification.

Reviewer 2 Report

The authors evaluated the pro-apoptotic effect of the COX2 inhibitor Previcox both in vivo and in vitro on canine mammary cancer. The work is original and can be published but it requires an important reorganization of the text. 

Line 1: the title doesn't reflect the content of the work; a pro-apoptotic effect has been demonstrated but there isn't a clear strong correlation with a better prognosis, even though the overall survival is available. It could be changed; furthermore, the word "preliminary" can be used because the case number is low.

Line 30: It's not clear the initial aim of the study because in the discussion you've written that the in vitro study came after the in vivo one (line 272) on a conceptual level; despite this fact, you presented all the work in an inverted way with the in vitro study first. You should change the discussion or the setup of the article. Furthermore, you should change the abstract conclusion because adjuvant ability has not been demonstrated.

line 47: The introduction lacks of information about the relation between Cox2 inhibition and apoptosis induction; can you add some of these information?

line 71: can you add some references about the Cox2 expression relationship with aggressive tumours in dogs and human? 

line 93: how may cells did you seed? What is the doubling time of each cell line? The culture confluence is very important for the MTT assay.

line 108: how may cells did you seed? How may replicates did you have for this analysis?

line 145: In the paragraph about the double immunostaining there aren't methodology information about the positive cells' count. Did you consider the whole nodule or just some fields? How many fields? At which magnification?

line 171: did you check the normality of the distributions? Considering that you know the overall survival, and the dog status I think, why didn't you perform the survival analysis (Kaplan Meier curves) associated with the treatment?  Furthermore, this type of data, even though the case number is low, can permit a multivariate analysis, did you check for the assumptions? The number of Cox2/caspase3 positive cells could be dependent on other variables. 

line 211: can you add the standard deviations in the table?

line 213: can you add in the table 2 the cox2/caspase3 count results for each case? Or if you want in a supplement table

line 272: as I mentioned before, this period doesn't fit very well with all the text because it is in contradiction with the structure of the article

can you add in the discussion some references about the mechanisms that bind cox2 with apoptosis? 

line 287: the results don't suggest that previcox is an adjuvant treatment but only that it triggers apoptosis in cox2 positive cells. Can you also add that the number of cases should be increase?

Author Response

Dear reviewer, thank you so much for you positive criticism for our manuscript improvement. We really appreciated you suggestion. We would like to highlight that our manuscript is a prospective study, including female dogs with triple negative mammary gland tumors. Performing a prospective study including patient from routine is really challenging and for this reason, we do not see a high number of prospective studies in veterinary medicine. To the best of our knowledge, this is the first prospective investigating the role of COX-2 expression on canine mammary gland tumors in vivo. For this reason, as first approach, we did not look for evaluating overall survival. Our main goal was investigate the pro apoptotic effect of Firocoxib in canine mammary gland tumor in vitro and in vivo. COX-2 expression in human cancer cells is associated with resistance to apoptosis. However, this is not disseminated knowledge in veterinary medicine, and although clinical benefits of COX-2 inhibitors is widely evaluated in canine cancers, no previous research was focused on this mechanism. Therefore, our research brings a unique perspective for understanding its antitumor effect. Please, find below the comment-by-comment of your recommendations.

Line 1: the title doesn't reflect the content of the work; a pro-apoptotic effect has been demonstrated but there isn't a clear strong correlation with a better prognosis, even though the overall survival is available. It could be changed; furthermore, the word "preliminary" can be used because the case number is low.

Answer: Dear reviewer, thank you so much for your comment. In our paper, we did not associate COX-2 with prognosis. We did not perform this correlation because the number of dogs is low. We aimed to evaluate the mechanism of Firocoxib (if induce apoptosis). For this reason, we did not use prognosis in the title.

Line 30: It's not clear the initial aim of the study because in the discussion you've written that the in vitro study came after the in vivo one (line 272) on a conceptual level; despite this fact, you presented all the work in an inverted way with the in vitro study first. You should change the discussion or the setup of the article. Furthermore, you should change the abstract conclusion because adjuvant ability has not been demonstrated.

Answer: Dear reviewer, our aim was to evaluate the pro apoptotic effect of firocoxib on canine mammary cancer. In our in vivo study, we investigate the pro apoptotic effect of firocoxib as a neoadjuvant treatment not adjuvant. The reviewer, is totally, right. We are sorry for this mistake. Then, we have change for neoadjuvant.

line 47: The introduction lacks of information about the relation between Cox2 inhibition and apoptosis induction; can you add some of these information?

Answer: Thank you so much for this kind comment. We have included the information as requested.

line 71: can you add some references about the Cox2 expression relationship with aggressive tumours in dogs and human? 

Answer: Thank you for your suggestion, we have added new references. Once again, thank you so much.

line 93: how may cells did you seed? What is the doubling time of each cell line? The culture confluence is very important for the MTT assay.

Answer: Dear reviewer, we have seeded 10.000 cells and the doubling time for UNESP-CM5 is around 10 hours and for UNESP-MM1 is around 24 hours. Therefore, we considered a confluency of 60% for the MTT analysis.

line 108: how may cells did you seed? How may replicates did you have for this analysis?

Answer: we have seeded 10.000 cells and experiments in technical and biological triplicate.

line 145: In the paragraph about the double immunostaining there aren't methodology information about the positive cells' count. Did you consider the whole nodule or just some fields? How many fields? At which magnification?

Answer: dear reviewer, we are so sorry that this information was lacking in the previous version. We have count a total of 1000 cells in high power field and have used to total number of double positive cells. We considered the hot spots.  We provided a more detailed information in the manuscript.

line 171: did you check the normality of the distributions? Considering that you know the overall survival, and the dog status I think, why didn't you perform the survival analysis (Kaplan Meier curves) associated with the treatment?  Furthermore, this type of data, even though the case number is low, can permit a multivariate analysis, did you check for the assumptions? The number of Cox2/caspase3 positive cells could be dependent on other variables. 

Answer: Dear reviewer, in this manuscript, we did not aimed to evaluate the association of COX-2 expression with survival as a prognostic factor. We performed a power analysis to ensure the reliability of our set of samples and 22 samples were the minimum sample size. To evaluate overall survival, our power analysis revealed a minimum number of 100 female dogs. For this reason, we did not include survival (since we will not be a reliable data). Doing a survival analysis, direct our manuscript for a side that we unfortunately can explore, since our sample size is low. Then, we decided that will be a weak analysis. It is important to highlight that in our case, each patient was control of himself (comparison of D0 versus D14). This type of approach minimize the assumption of other variables modifying the results.

line 211: can you add the standard deviations in the table?

Answer: Dear reviewer, this data provide a total of events in flow cytometry. Therefore, if the percentage of cell positive or negative in each cell line for each marker, considering the counting of 10.000 cells. Could you please be more specific regarding the stardart deviations for each cell? We are really sorry, but we did not understand. 

line 213: can you add in the table 2 the cox2/caspase3 count results for each case? Or if you want in a supplement table.

Answer: Yes, it will be a pleasure to provide this data. Please, find in the new supplementary table 1.

line 272: as I mentioned before, this period doesn't fit very well with all the text because it is in contradiction with the structure of the article

Answer: Dear reviewer, we are so sorry, but we did not understand the comment. We first performed the treatment of cancer cell lines, to have an in vitro evidence of apoptosis in mammary cancer cell lines, to further investigate in vivo. The connection of both analysis, was associated with a primary evidence of apoptosis in vitro to try in vivo.

can you add in the discussion some references about the mechanisms that bind cox2 with apoptosis? 

Answer: Thank you for the suggestion. In human cancers, this role is still under debate. However, is well-established that COX-2 expression by cancer cells is associated with resistance to apoptosis. Therefore, blocking the COX-2 enzyme can revert cancer cells resistance and enhance chemotherapy response. We have added some information in the discussion.

line 287: the results don't suggest that previcox is an adjuvant treatment but only that it triggers apoptosis in cox2 positive cells. Can you also add that the number of cases should be increase?

Answer: we have included the information, as suggested.

Round 2

Reviewer 2 Report

The authors have made the requested changes.